# Raman spectroscopic signature of fractionalized excitations in the harmonic-honeycomb iridates $\beta$- and $\gamma$-Li$_2$IrO$_3$

A. Glamazda[1], P. Lemmens[2,3], S.-H. Do[1], Y.S. Choi[1] & K.-Y. Choi[1]

The fractionalization of elementary excitations in quantum spin systems is a central theme in current condensed matter physics. The Kitaev honeycomb spin model provides a prominent example of exotic fractionalized quasiparticles, composed of itinerant Majorana fermions and gapped gauge fluxes. However, identification of the Majorana fermions in a three-dimensional honeycomb lattice remains elusive. Here we report spectroscopic signatures of fractional excitations in the harmonic-honeycomb iridates $\beta$- and $\gamma$-Li$_2$IrO$_3$. Using polarization-resolved Raman spectroscopy, we find that the dynamical Raman response of $\beta$- and $\gamma$-Li$_2$IrO$_3$ features a broad scattering continuum with distinct polarization and composition dependence. The temperature dependence of the Raman spectral weight is dominated by the thermal damping of fermionic excitations. These results suggest the emergence of Majorana fermions from spin fractionalization in a three-dimensional Kitaev–Heisenberg system.

[1] Department of Physics, Chung-Ang University, 84 Heukseok-Ro, Dongjak-Gu, Seoul 156-756, Republic of Korea. [2] Institute for Condensed Matter Physics, TU Braunschweig, D-38106 Braunschweig, Germany. [3] Laboratory for Emerging Nanometrology (LENA), TU Braunschweig, D-38106 Braunschweig, Germany. Correspondence and requests for materials should be addressed to K.-Y.C. (email: kchoi@cau.ac.kr).

The fractionalization of elementary excitations is a characteristic feature of quantum spin liquids. Such a liquid evades conventional magnetic order even at $T = 0$ K and thereby preserves all symmetries of the underlying spin Hamiltonian. In the last decade, there has been significant progress in the experimental identification of quantum spin liquids in a class of geometrically frustrated Heisenberg magnets[1] with elementary excitations that are given by chargeless spinons carrying spin $s = 1/2$. For two-dimensional (2D) triangular and Kagome lattices, however, a quantitative understanding of spinons remains unsatisfactory due to a lack of reliable theoretical methods of handling macroscopic degenerate ground states[2,3].

In this context, the exactly solvable Kitaev honeycomb model offers a genuine opportunity of exploring spin liquid physics on a more quantitative level as the spin response functions for spin liquids can be analytically computed[4–9]. Until now, searching for Kitaev materials has been centred on the iridates $\alpha$-$A_2IrO_3$ ($A$ = Li, Na) and the ruthenates $\alpha$-$RuCl_3$, in which $Ir^{4+}$ ($5d^5$) or $Ru^{3+}$ ($4d^5$) ions create the $J_{eff} = 1/2$ Mott state by the combined effects of strong spin–orbit coupling, electronic correlations and crystal field[10–19]. In the tricoordinated geometry of edge-sharing $IrO_6$ or $RuO_6$ octahedra, $J_{eff} = 1/2$ moments interact via two 90° Ir-O-Ir exchange paths, giving rise to anisotropic bond-dependent Kitaev interactions[20,21].

A potential drawback is the development of long-range order at low temperatures and the presence of Heisenberg interactions in real materials[22]. Despite the detrimental effects of residual interactions, $\alpha$-$RuCl_3$ shows an indication of the spin fractionalization through a continuum-like excitation in the Raman response and a high-energy Majorana excitation in inelastic neutron scattering[15,17]. In contrast, only a subtle signature of Kitaev interactions exists for $\alpha$-$A_2IrO_3$. The absence of well-defined fractionalized excitations in the iridates is ascribed to the structural distortion of planar Ir-O-Ir bonds[10].

In search for a new platform for Kitaev magnetism, the harmonic series of hyperhoneycomb lattices, $\beta-$ and $\gamma-Li_2IrO_3$, were discovered[23,24]. These structural polytypes have the same tricoordinated network of Ir ions as the layered $\alpha$-$A_2IrO_3$, and thus are a three-dimensional (3D) analogue of the honeycomb iridate materials. An ensuing question is whether quantum spin liquids are preserved in such a 3D generalization of the Kitaev model[25–29].

Specific heat and magnetic susceptibility data evidence long-range magnetic order at $T_N = 38$ and 39.5 K for $\beta$- and $\gamma$-$Li_2IrO_3$, respectively[23,24]. Strikingly, the two distinct structural polytypes display a similar incommensurate spiral order with non-coplanar and counter-rotating moments. This implies that Kitaev interactions dictate magnetism of both compounds[30–33]. In this respect, these materials present promising candidates for attesting the elusive spin fractionalization in the 3D honeycomb lattice. Currently, it is difficult to detect Majorana fermions with X-ray and neutron-scattering technique, as only submillimetre-sized crystals are available. Therefore, Raman spectroscopy is the most suitable method for addressing this issue, because it directly probes Majorana fermion density of states. Moreover, detailed theoretical predictions of the polarization dependence of magnetic Raman scattering in the hyperhoneycomb lattice exist that can prove fractionalized fermionic excitations[8].

In this study, we provide Raman spectroscopic evidence for weakly confined Majorana fermions in 3D honeycomb iridate materials. The polarization and composition dependence of broad spinon continua point towards a different topology of spinon bands comparing $\beta$- and $\gamma$-$Li_2IrO_3$. In addition, the temperature dependence of the integrated Raman intensity obeys the Fermi statistics, being in stark contrast to bosonic Raman spectra observed in conventional insulating magnets. These results demonstrate the emergence of fermionic excitations from the spin fractionalization in a 3D honeycomb lattice.

## Results

**Polarization dependence of Raman spectra.** Figure 1a shows the crystal structures of $\beta$- and $\gamma$-$Li_2IrO_3$. $\beta$-$Li_2IrO_3$ consists of the zigzag chains (blue and orange sticks), which alternate in orientation between the two basal plane diagonals and are connected via the bridging bonds (green stick) along the $c$ axis. In $\gamma$-$Li_2IrO_3$, two interlaced honeycomb layers alternate along the $c$ axis. Figure 1b,c presents the polarization-dependent Raman responses $\chi''(\omega)$ of $\beta$- and $\gamma$-$Li_2IrO_3$ measured at $T = 6$ K in two different scattering geometries. Here the notation $(xy)$ with $x = a$ and $y = b, c$ refers to the incident and scattered light polarizations, which are parallel to the crystalline $x$ and $y$ axis, respectively. $\chi''(\omega)$ presents the dynamical properties of collective excitations and is obtained from the raw Raman spectra $I(\omega)$ using the relation $I(\omega) \propto [1 + n(\omega)]\chi''(\omega)$ where $1 + n(\omega) = 1/\left(1 - e^{-\hbar\omega/k_B T}\right)$ is the Bose thermal factor.

Within the Fleury–Loudon–Elliott theory[34], the magnetic Raman scattering intensity of a 3D Kitaev system is given by the density of states of a weighted two-Majorana spinon, $I(\omega) = \pi \sum_{m,n;k} \delta(\omega - \varepsilon_{m,k} - \varepsilon_{n,k})|B_{mn,k}|^2$, where $\varepsilon_{m,k}$ is a Majorana spinon dispersion with the band indices $m$, $n = 1,2(1,2,3)$ for $\beta(\gamma)$-$Li_2IrO_3$ and $B_{mn,k}$ is the matrix element creating two Majorana excitations[8]. The observed $\chi''(\omega)$ is composed of sharp phonon excitations superimposed on a broad, featureless continuum extending up to 200 meV. The Raman-active phonon modes are presented in Supplementary Fig. 1 and Supplementary Tables 1 and 2 (see also Supplementary Note 1 for details). The magnetic continuum arises mainly from two-Majorana spinon excitations. This assignment is analogue to observations in the 2D honeycomb lattice $\alpha$-$RuCl_3$, in which a broad continuum is taken as evidence of fractionalized excitations[15]. The striking similarity of the magnetic response between $\alpha$-$RuCl_3$ and $\beta$- and $\gamma$-$Li_2IrO_3$ suggests that the 3D honeycomb iridates and the 2D honeycomb ruthenate realize Kitaev magnetism to a similar extent.

Thanks to the multiple spinon bands in the 3D harmonic honeycomb system, the Raman response of $\beta$- and $\gamma$-$Li_2IrO_3$ will be polarization- and composition-dependent, emulating a number of band edges and van Hove singularities[8]. As seen in Fig. 1b,c, the iridate compounds show commonly an asymmetric magnetic response towards lower energy. The polarization dependence is mostly evident in the $\omega$-dependence of $\chi''(\omega)$. Compared with $\chi''(ac)$, $\chi''(ab)$ with green shading becomes systematically suppressed as $\omega \to 0$. Examining its composition dependence, $\chi''(ac)$ of $\beta$- and $\gamma$-$Li_2IrO_3$ is plotted together in Fig. 1d after subtracting phonon modes. $\chi''(ac)$ of $\beta$-$Li_2IrO_3$ shows a round maximum at $\sim 33$ meV, whereas its spectral weight is depressed to zero as $\omega \to 0$. In contrast, $\chi''(ac)$ of $\gamma$-$Li_2IrO_3$ has two maxima at 26 and 102 meV along with a finite excitation gap of $\Delta = 5 - 6$ meV marked by the arrows in Fig. 1c,d. Here, the extracted gap is estimated by a linear extrapolation of $\chi''(\omega)$. The slightly richer spectrum of $\gamma$-$Li_2IrO_3$ than $\beta$-$Li_2IrO_3$ is linked to the increasing number of Majorana spinon bands. Thus, these results establish a subtle yet discernible polarization and composition dependence of $\chi''(\omega)$ in the 3D hyperhoneycomb compounds.

A related question is to what extent the hyperhoneycomb iridate materials retain the characteristic of Majorana fermions inherent to the 3D Kitaev model. For this purpose, we first compare the experimental and theoretical Raman response of $\beta$-$Li_2IrO_3$, which lies at the near-isotropic point with $J^x = J^y \approx J^z$

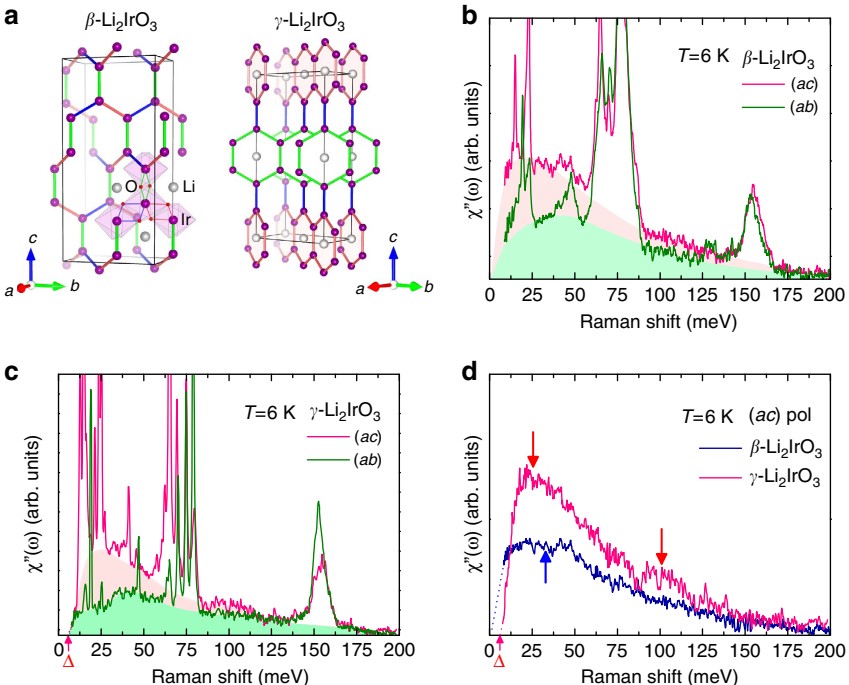

**Figure 1 | Crystal structure of $\beta$- and $\gamma$-Li$_2$IrO$_3$ and their Raman responses $\chi''(\omega)$ in two different scattering channels.** (**a**) Hyperhoneycomb lattice in $\beta$- and $\gamma$-Li$_2$IrO$_3$. Purple, red and grey balls are iridium, oxygen and lithium atoms, respectively. In $\beta$-Li$_2$IrO$_3$, the alternating blue and orange sticks depict the twisted zigzag chains and the green sticks are the bond connecting the zigzag chains. In $\gamma$-Li$_2$IrO$_3$, two iridium hexagons are arranged in an alternating way along the $c$ axis. (**b**,**c**) Polarization dependence of the Raman response $\chi''(\omega)$ of $\beta$- and $\gamma$-Li$_2$IrO$_3$ in (*ac*) and (*ab*) scattering channels measured at $T = 6$ K. A magnetic continuum in (*ab*) polarization is painted with green shading. An additional magnetic excitation seen in (*ac*) polarization is highlighted with incarnadine shading. (**d**) Comparison of the Raman responses $\chi''(\omega)$ between $\beta$- and $\gamma$-Li$_2$IrO$_3$ in the (*ac*) scattering channel after subtracting phonon peaks. The arrows mark the local maximum of the spectral weight and the $\Delta$ symbol indicates an energy gap.

(see Supplementary Fig. 2 and Supplementary Note 2 for details). Similar trends are observed in the polarization dependence of the scattering intensity; the (*ac*) polarization spectrum has a much stronger intensity than the (*ab*) polarization spectrum, being in line with the theoretical calculations[8]. However, the low-energy spectrum does not open an excitation gap in the (*ab*) scattering channel and the fine spectral features anticipated in the bare two-Majorana spinon density of states do not show up. There is not much difference in the polarization dependence for the case of $\gamma$-Li$_2$IrO$_3$, which possesses three Majorana spinon bands and is at the anisotropic point with $J^x \neq J^y \neq J^z$ (see Supplementary Fig. 2 and Supplementary Note 2 for the local bond geometry). The absence of the sharp spectral features and polarization-dependent spectral widths is ascribed to the unwanted spin-exchange terms including Heisenberg, off-diagonal and longer-range interactions. These subdominant terms on the one hand lead to a weak confinement of Majorana spinons, rendering the smearing out of the van-Hove singularities and the softening of spectral weight. On the other hand, they give rise to a bosonic (magnon) contribution to the magnetic continuum at low energies. In this regard, the excitation gap in $\gamma$-Li$_2$IrO$_3$ corresponds to an energy gap in the low-energy spin waves. As the pseudospin $s = 1/2$ has a negligible single ion anisotropy, the anisotropic Kitaev interactions of $\gamma$-Li$_2$IrO$_3$ are responsible for opening the large gap. Notably, no obvious energy gap is present in the low-energy excitations of $\beta$-Li$_2$IrO$_3$ with nearly isotropic Kitaev interactions.

Before proceeding, we estimate the Kitaev exchange interaction $J_z = 17$ meV from the upper cutoff energy of the magnetic continuum. The extracted value is almost two times bigger than $J_z = 8$ meV of $\alpha$-RuCl$_3$ (ref. 15), being consistent with larger spatial extent of Ir orbitals.

**Evolution of fermionic excitations**. The temperature dependence of the Raman spectra was measured for both $\beta$- and $\gamma$-Li$_2$IrO$_3$ in the (*cc*) and (*ac*) scattering symmetries, respectively. The representative spectra are shown in Fig. 2a,b. The broad magnetic continuum marked with pink shading develops progressively into a quasi-elastic response at low energies on heating through $T_N$. The low-energy magnetic scattering grows more rapidly in $\beta$- than $\gamma$-Li$_2$IrO$_3$, because the latter has the large excitation gap. The magnetic Raman scattering at finite temperatures arises from dynamical spin fluctuations in a quantum paramagnetic state and can provide a good measure of the thermal fractionalization of quantum spins. The integrated Raman intensity in the energy range of $1.5 J_z < \hbar\omega < 3 J_z$ is plotted as a function of temperature in Fig. 2c,d. The temperature dependence of the integrated $I(\omega)$ is well fitted by a sum of the Bose and the two-fermion scattering contribution $(1 - f(\omega))^2$ with the Fermi distribution function $f(\omega) = 1/(1 + e^{\hbar\omega/k_B T})$ (ref. 35). The Bose contribution describes bosonic excitations such as magnons, whereas the two-fermion contribution is related to the creation or annihilation of pairs of fermions. The deduced energy $\hbar\omega = 0.76 - 79 J_z$ of fermions for $\beta$- and $\gamma$-Li$_2$IrO$_3$ validates the fitting procedure adopting a Fermi distribution function. Here we stress that the thermal fluctuations of fractionalized fermionic excitations are a Raman spectroscopic evidence of proximity to a Kitaev spin liquid. Essentially the same fermionic excitations were inferred from the $T$-dependence of the integrated spectral weight in $\alpha$-RuCl$_3$ (ref. 35).

Figure 2e,f shows the Raman conductivity $\chi''(\omega)/\omega$ versus temperature. The Raman conductivity features a pronounced peak centred at $\omega = 0$. The low-energy Raman response exhibits a strong enhancement with increasing temperature. The intermediate-to-high energy $\chi''(\omega)/\omega$ above 30 meV dampens hardly with temperature. From the Raman conductivity we can

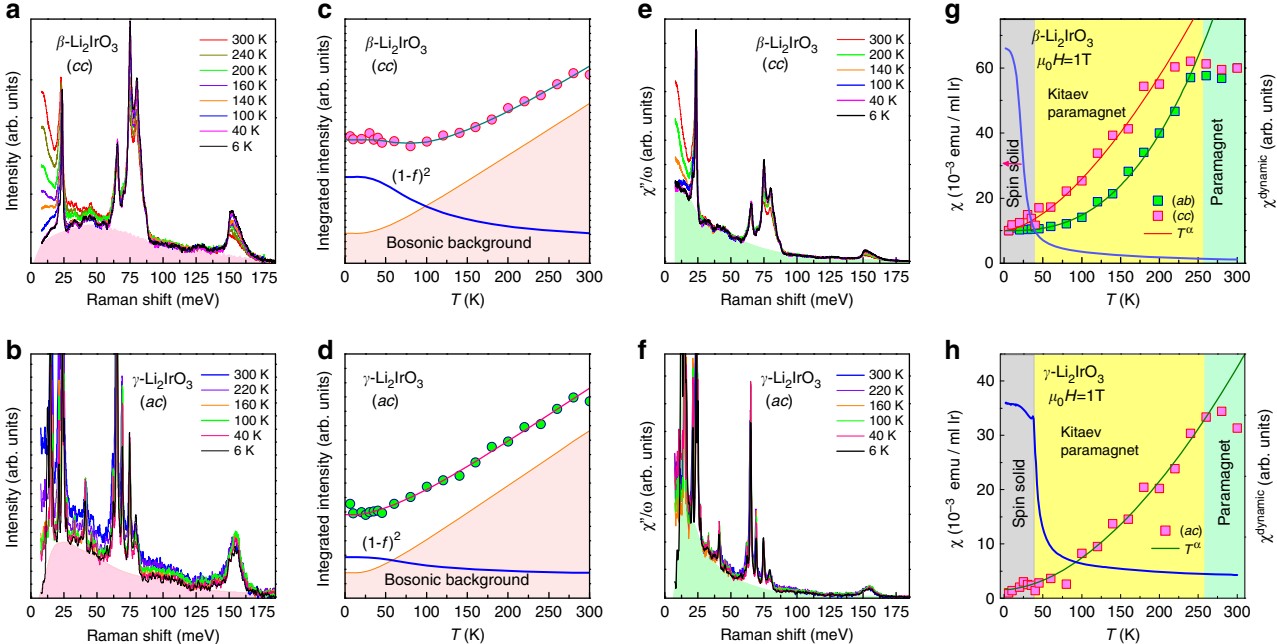

**Figure 2 | Raman spectra and Raman conductivity $\chi''(\omega)/\omega$ of $\beta$- and $\gamma$-Li$_2$IrO$_3$ as a function of temperature.** (**a,b**) Temperature dependence of the Raman spectrum measured in (*cc*) polarization for $\beta$-Li$_2$IrO$_3$ and in (*ac*) polarization for $\gamma$-Li$_2$IrO$_3$. (**c,d**) Temperature dependence of the integrated Raman intensity obtained in the energy window from 25 to 51 meV. The shading area indicates the bosonic background and the solid lines are a fit to the two-fermion creation or annihilation process, $(1 - f(\omega))^2$ with the Fermi distribution function $f(\omega) = 1/(1 + e^{\hbar\omega/k_B T})$. (**e,f**) Temperature dependence of the Raman conductivity $\chi''(\omega)/\omega$ in (*cc*) polarization for $\beta$-Li$_2$IrO$_3$ and in (*ac*) polarization for $\gamma$-Li$_2$IrO$_3$. The green shadings are a magnetic continuum. (**g,h**) Temperature dependence of the dynamic Raman susceptibility deduced from the Kramers Kronig relation. Temperature dependence of the static spin susceptibility is plotted together for comparison. The solid lines are a power-law fit to the data, $\chi^{dyn}(T) \sim T^\alpha$.

define a dynamic Raman susceptibility using Kramers–Kronig relation $\chi^{dyn} = lim_{\omega \to 0} \chi(k = 0, \omega) \equiv \frac{2}{\pi} \int_0^\infty \frac{\chi''(\omega)}{\omega} d\omega$, that is, by first extrapolating the data from the lowest energy measured down to 0 meV and then integrating up to 200 meV. It is noteworthy to mention that $\chi^{dyn}$ is in the dynamic limit of $\chi^{static} = lim_{k \to 0} \chi(k, \omega = 0)$[36]. Figure 2g,h plots the temperature dependence of $\chi^{dyn}(T)$ of $\beta$- and $\gamma$-Li$_2$IrO$_3$. Irrespective of polarization and composition, $\chi^{dyn}(T)$ shows a similar variation with temperature. On heating above $T_N$, $\chi^{dyn}(T)$ increases rapidly and then saturates for temperatures above $T^* = 220 - 260$ K. Remarkably, the energy corresponding to $T^*$ is comparable to the Kitaev exchange interaction of $J_z = 17$ meV. We further note that the 2D Heisenberg–Kitaev material $\alpha$-RuCl$_3$ exhibits also a drastic change of magnetic dynamics through $T \sim J_z = 100$–140 K (ref. 15). For temperatures below $T^*$, the power law gives a reasonable description of $\chi^{dyn}(T) \sim T^\alpha$ with $\alpha = 1.58 \pm 0.05$ and $2.64 \pm 0.09$ in the respective (*cc*) and (*ab*) polarization for $\beta$-Li$_2$IrO$_3$ and $\alpha = 1.77 \pm 0.06$ for $\gamma$-Li$_2$IrO$_3$. As discussed in Supplementary Fig. 3 and Supplementary Note 3, $\chi^{dyn}(T)$ is temperature independent in the paramagnetic phase as paramagnetic spins are uncorrelated. This is contrasted to the power-law dependence of $\chi^{dyn}(T)$ in a spin liquid. This power-law is associated with slowly decaying correlations inherent to a spin liquid[37] and the onset temperature $T^*$ heralds a thermal fractionalization of Kitaev spins[9].

We now compare the dynamic Raman susceptibility with the static spin susceptibility given by SQUID magnetometry. As evident from Fig. 2g,h, they behave in an opposite way. This discrepancy indicates that a large number of correlated spins are present in the limit $\omega \to 0$.

**Fano resonance of optical phonon and magnetic specific heat.** The phonon Raman spectra unveil a strongly asymmetry lineshape

at 24 meV in $\beta$-Li$_2$IrO$_3$ (see Fig. 3a) that is well fitted by a Fano profile $I(\omega) = I_0(q + \varepsilon)^2/(1 + \varepsilon^2)$ (ref. 38). The reduced energy is defined by $\varepsilon = (\omega - \omega_0)/\Gamma$ where $\omega_0$ is the bare phonon frequency, $\Gamma$ the linewidth and $q$ the asymmetry parameter. In Fig. 3b,c, we plot the resulting frequency shift, the linewidth and the Fano asymmetry as a function of temperature. The errors are within a symbol size. Based on lattice dynamical calculations (see Supplementary Note 1), this phonon is assigned to an $A_g$ symmetry mode, which involves contracting vibrations of Ir atoms along the $c$ axis (see the sketch in the inset of Fig. 3a). Therefore, the observed anomalies could shed some light on the thermal evolution of Kitaev physics, because a Fano resonance has its root in strong coupling of phonons to a continuum of excitations.

With decreasing temperature, the Fano asymmetry, $1/|q|$, increases continuously and then becomes constant below the magnetic ordering temperature. As clearly seen from Fig. 3b, the temperature dependence of $1/|q|$ follows the two-fermion scattering form $(1 - f(\omega))^2$, which gives a nice description of the temperature dependence of the integrated $I(\omega)$ (see Fig. 2c,d). It is striking that the magnitude of the Fano asymmetry parallels a thermal damping of the fermionic excitations. In a Kitaev honeycomb system, spins are thermally fractionalized into the itinerant Majorana spinons[9]. As a result, the continuum stemming from the spin fractionalization strongly couples to lattice vibrations that mediate the Kitaev interaction. It is noteworthy that the 24 meV mode involves the contracting motion of the bridging bonds between consecutive zigzag chains along the $c$ axis. In addition, $\alpha$-RuCl$_3$ shows a Fano resonance of a phonon, which reinforces our assertion that the Fano asymmetry is an indicator of the thermal fractionalization of spins into the Majorana fermions[15].

As the temperature is lowered, phonon modes usually increase in energy and narrow in linewidth due to a suppression of anharmonic phonon–phonon interactions. Indeed, as shown in

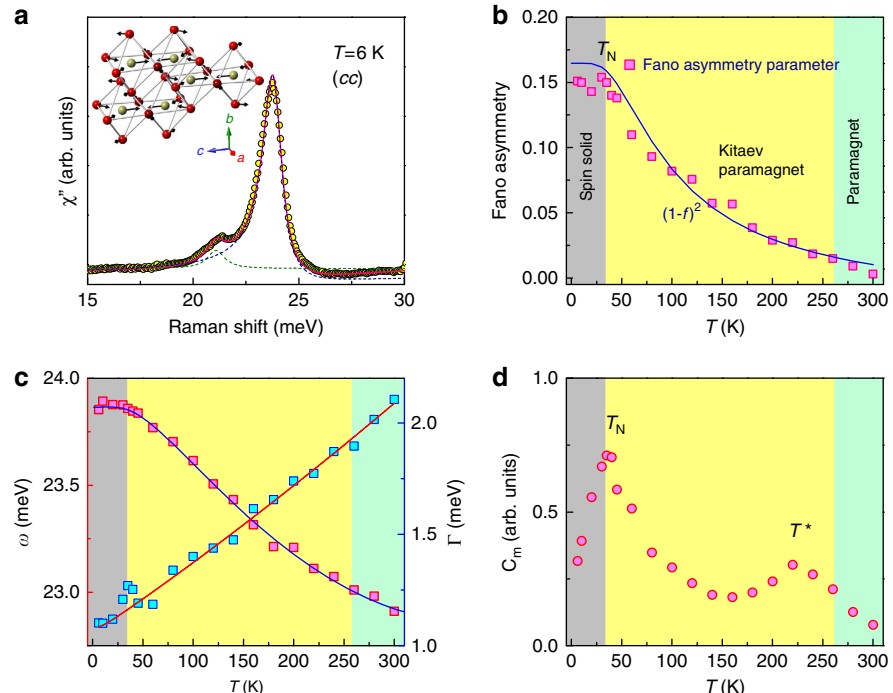

**Figure 3 | A Fano resonance of the 24 meV phonon mode and magnetic specific heat.** (**a**) Fit of 24 meV phonon to a Fano profile after subtracting a temperature-dependent magnetic background. The 21 meV phonon on a low-energy side of the Fano resonance is fitted together with a Lorentzian profile. The inset depicts a schematic representation of eigenvector of the 24 meV $A_g$ symmetry mode. The amplitude of the vibrations is represented by the arrow length. Golden balls indicate Ir ions and red balls are O ions. The Li atoms are omitted for simplicity. (**b**) Temperature dependence of the Fano asymmetry $1/|q|$ plotted together with the two-fermion form $(1-f(\omega))^2$ (solid line). (**c**) The energy $\omega$ and linewidth $\Gamma$ as a function of temperature. The solid lines are a fit to an anharmonic phonon model. (**d**) Temperature dependence of the magnetic specific heat $C_m$ derived from the Raman conductivity $\chi''(\omega)/\omega$.

Fig. 3c, the temperature dependence of $\omega$ and $\Gamma$ is well described by conventional anharmonic decay processes (see also Supplementary Note 4). A small kink in $\Gamma$ occurs at the onset temperature of the magnetic ordering. Unlike the 2D honeycomb lattice $\alpha$-RuCl$_3$ (ref. 15), however, there appears to be no noticeable renormalization of the phonon energy and linewidth on crossing $T_N$ and $T^\star$. This may be due to the large unit cell of the 3D network of spins and low crystal symmetry. In such a complex spin network, lattice vibrations involve simultaneous modulations of different magnetic exchange paths and thus spin correlation effects on the phonon are largely nullified. This scenario is supported by the lacking Fano resonance in $\gamma$-Li$_2$IrO$_3$ having a lower symmetry and stronger trigonal distortion compared with $\beta$-Li$_2$IrO$_3$.

Next, we turn to the magnetic-specific heat $C_m$ of a Kitaev system. Spin fractionalization into two types of the Majorana fermions leads to a two-peak structure[9] and a rich phenomenology in its temperature dependence. A high-$T$ crossover is driven by the itinerant Majorana fermions and linked to the development of short-range correlations between the nearest-neighbour spins. A low-$T$ topological transition is expected due to the $Z_2$ fluxes. Quasielastic Raman scattering can be used to derive the magnetic specific heat using a hydrodynamic limit of the spin correlation function[39]. Next, the Raman conductivity is associated with $C_m$ by the relation $\chi''(\omega)/\omega \propto C_m T I_L(\omega)$, where $I_L(\omega)$ is the Lorentzian spectral function (see the Methods for details)[40–42]. A fit to this equation allows evaluating $C_m(T)$ from the integration of $\chi''(\omega)/\omega$ scaled by $T$. In Fig. 3d, the resulting $C_m$ versus $T$ is plotted. We confirm the two peaks at $T_N = 0.1\,J$ and $T^\star \sim J$. The high-$T$ peak at $T^\star \sim J$ is somewhat higher than that of the theoretical value of $0.6\,J$ (ref. 9). In addition, the predicted topological transition at

$T \sim 0.005\,J$ is pre-empted by the long-range magnetic order at $T_N = 0.1\,J$. We ascribe the discrepancy between experiment and theory to residual interactions, which lift the Raman selection rules of probing the Majorana fermions. In spite of the magnetic order, the persistent two-peak structure in $C_m$ suggests that the hyperhoneycomb iridates are in proximity to a Kitaev spin liquid phase.

## Discussion

Having established that $\beta$- and $\gamma$-Li$_2$IrO$_3$ have fractionalized fermionic excitations, it is due to compare them with spinon excitations in the well-characterized kagome Heisenberg antiferromagnet ZnCu$_3$(OH)$_6$Cl$_2$ (refs 3,43). In such a system, geometrical frustration is the key element.

Despite disparate sources of fractionalized excitations, a number of key features in the spectral shape and temperature dependence of magnetic scattering, as well as in the Fano (anti)resonance of optical phonons (see the asterisks in Fig. 4) are common to $\beta$-Li$_2$IrO$_3$ and ZnCu$_3$(OH)$_6$Cl$_2$. Both compounds show a broad continuum with a rounded maximum at low energies, as shown in Fig. 4. In ZnCu$_3$(OH)$_6$Cl$_2$, the low-energy response decreases linearly down to zero frequency and the magnetic continuum extends up to a high-energy cutoff at $6J$ with $J \approx 16$ meV. The former property suggests the formation of a gapless spin liquid and the latter the existence of multiple spinon scattering processes[43,44]. In a similar manner, the low-energy spectral weight of $\beta$-Li$_2$IrO$_3$ drops to zero with a steeper slope. The similar behaviour observed in the two compounds with different spin and lattice topologies may be due the fact that the bare spinon density of states is modified due to Dzyaloshinskii–Moriya interactions and antisite disorder in

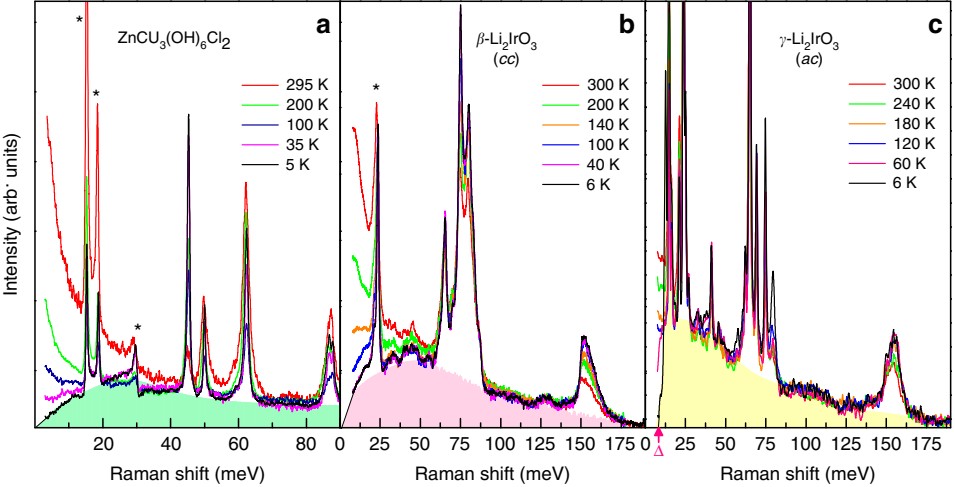

**Figure 4 | Comparison of the spinon continuum between the kagome and the honeycomb lattices. (a)** Raman spectra of $ZnCu_3(OH)_6Cl_2$ in ($aa$) polarization as a function of temperature. The data are taken from ref. 43. The shadings are the spinon continuum. The asterisks denote optical phonons with Fano line shape. (**b,c**) Raman spectra of $\beta$- and $\gamma$-Li$_2$IrO$_3$ at various temperatures.

$ZnCu_3(OH)_6Cl_2$ and other spin-exchange interactions in $\beta$-Li$_2$IrO$_3$. The resemblance becomes less clear for $\gamma$-Li$_2$IrO$_3$, mainly because the anisotropic Kitaev exchange interactions open a large excitation gap in the low-energy excitations.

Next, we discuss the temperature dependence of the magnetic continuum. Irrespective of the spinon topology and spin-exchange type, the three studied compounds share essentially the same phenomenology. The key feature is the evolution of a spinon continuum into a quasi-elastic response with increasing temperature. This is well characterized by the power-law dependence of $\chi^{dyn}(T) \sim T^{\alpha}$. The exponent of $\alpha = 1.58 - 2.64$ is not much different comparing the three compounds[43]. As discussed in Supplementary Note 3, this power-law behaviour is inherent to a long-range entangled spin liquid and is completely different from what is expected for conventional magnets. Despite the distinct spinon band structure, the spinon correlations may be not very different between the 2D kagome and the 3D hyperhoneycomb lattice.

The last remark concerns that $\chi^{dyn}(T)$ of the 3D hyper-honeycomb materials starts to deviate from a power-law behaviour at 220 K. At the respective temperature, the magnetic specific heat shows as a broad peak identified as a thermal crossover from a paramagnet to a Kitaev paramagnet. This anomaly is absent in $ZnCu_3(OH)_6Cl_2$ with a single type of spinon and thus unique to $\beta$- and $\gamma$-Li$_2$IrO$_3$ having two species of Majorana fermions.

In summary, a Raman scattering study of the 3D honeycomb materials $\beta$- and $\gamma$-Li$_2$IrO$_3$ provides evidence for the presence of Majorana fermionic excitations. A polarization, temperature and composition dependence of a magnetic continuum indicates a distinct topology of spinon bands between $\beta$- and $\gamma$-Li$_2$IrO$_3$. The temperature dependence of an integrated Raman response and the two-peak structure in specific heat demonstrate that a thermal fractionalization of spins brings about fermionic excitations and that the 3D harmonic-honeycomb iridates realize proximate spin liquid at elevated temperatures. These results expand the concept of fractionalized quasiparticles to a 3D Kitaev–Heisenberg spin system.

## Methods

**Samples.** Single crystals of $\beta$-Li$_2$IrO$_3$ were grown by a flux method. The starting materials Li$_2$CO$_3$, IrO$_2$ and LiCl with ratio 10:1:100 were mixed together and pressed into a pellet. The pellet was placed in an alumina crucible, heated to 1,100 °C for 24 h and then cooled down to 700 °C for 14 h. Black powder-like

crystal grains appeared at a bottom of the crucible. The collected grains were washed by distilled water for the removal of the LiCl flux and filtered. The obtained crystals are of a size of $30 - 50 \mu m$. To grow single crystals of $\gamma$-Li$_2$IrO$_3$, polycrystalline pellets of $\alpha$-Li$_2$IrO$_3$ were first prepared. The prepared $\alpha$-phase pellet was heated to 1,170 °C for 72 h and slowly cooled down to 900 °C in air. Shinny black crystals with a size of 100 $\mu m$ were obtained on the surface of the pellet. The phase purity and composition of $\beta$- and $\gamma$-Li$_2$IrO$_3$ were confirmed via powder X-ray diffraction. Their bulk magnetic susceptibility is presented in Fig. 2g,h of the main text.

**Raman scattering experiment.** A polarized, resolved Raman spectroscopy was employed to detect spin and phonon excitations of single crystals of $\beta$- and $\gamma$-Li$_2$IrO$_3$. Raman scattering experiments were performed in backscattering geometry with the excitation line $\lambda = 532.1$ nm of a Nd:YAG (neodymium-doped yttrium aluminium garnet) solid-state laser. The scattered spectra were collected using a micro-Raman spectrometer (Jobin Yvon LabRam) equipped with a liquid-nitrogen-cooled charge-coupled device. A notch filter and a dielectric edge filter were used to reject Rayleigh scattering to a lower cutoff frequency of 60 cm$^{-1}$. The laser beam was focused to a few-micrometre-diameter spot on the surface of the crystal using a $\times 50$ magnification microscope objective. The samples were mounted onto a liquid-He-cooled continuous flow cryostat, while varying a temperature between 6 and 300 K. All Raman spectra were corrected for heating.

**Analysis of quasi-elastic Raman scattering.** Quasi-elastic light scattering arises from either diffusive fluctuations of a four-spin time correlation function or fluctuations of the magnetic energy density. According to Reiter[40] and Halley[41], a two-spin process leads to scattering intensity for temperatures above the critical temperature;

$$I(\omega) \propto \int_{-\infty}^{\infty} e^{-i\omega t} dt \langle E(k,t)E^*(k,0) \rangle, \qquad (1)$$

where $E(k,t)$ is a magnetic energy density given by the Fourier transform of $E(r) = -\langle \sum_{i>j} J_{ij} S_i \cdot S_j \delta(r - r_i) \rangle$ with the position of the $i$th spin $r_i$. Applying the fluctuation–dissipation theorem in the hydrodynamic limit[41], equation (1) is simplified to

$$I(\omega) \propto \frac{\omega}{1 - e^{-\beta \hbar \omega}} \frac{C_m T D k^2}{\omega^2 + (Dk^2)^2}, \qquad (2)$$

where $\beta = 1/k_B T$, $C_m$ is the magnetic specific heat and $D$ is the thermal diffusion constant $D = K/C_m$ with the magnetic contribution to the thermal conductivity $K$. Equation (2) can be rewritten in terms of a Raman susceptibility $\chi''(\omega)$,

$$\frac{\chi''(\omega)}{\omega} \propto C_m T \frac{Dk^2}{\omega^2 + (Dk^2)^2}. \qquad (3)$$

This relation is employed to extract the magnetic specific heat from the Raman conductivity in the main text.

**Data availability.** The authors declare that the data supporting the findings of this study are available within the article and its Supplementary Information files.

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

## Acknowledgements

This work was supported by the Korea Research Foundation (KRF) grant funded by the Korea government (MEST) (Grant Number 2009-0076079), as well as by German-Israeli Foundation (GIF, 1171-486 189.14/2011), the NTH-School Contacts in Nanosystems: Interactions, Control and Quantum Dynamics, the Braunschweig International Graduate School of Metrology and DFG-RTG 1952/1, Metrology for Complex Nanosystems.

## Author contributions

A.G. set up and carried out the Raman scattering measurements and analysed data. S.H.D. and Y.S.C. synthesized and characterized the samples. K.Y.C. and P.L. planned and coordinated the project. A.G., P.L. and K.Y.C. wrote the paper. All authors discussed the results and commented on the manuscript.

## Additional information

**Competing financial interests:** The authors declare no competing financial interests.

