## [Peer Review File · Nature Communications]

Reviewers' comments:

Reviewer #1 (Remarks to the Author):

The authors have addressed most of the points which were raised previously and now carefully state that they 'observe signatures of fractionalization' instead of 'signatures of a quantum spin liquid state'. The manuscript is very timely and contains important new results on both the beta and gamma Li₂IrO₃ which further establish these 3D materials as magnetic materials dominated by the celebrated Kitaev interaction. Their - arguably indirect - evidence for fractionalization will motivate further studies and will receive a lot of attention. The agreement between the clear Raman scattering predictions of Perreault et al. Ref.[8] for the 3D Kitaev spin liquid states is not great but nevertheless several features are surprisingly consistent. In particular the temperature dependence corroborates the authors interpretation in terms of fractionalized particles.

There are additional comments and points which the authors need to clarify, see below. Once these have been addressed, I do recommend publication of this important work in Nature Communication.

- On page 3, the authors write "We provide here Raman spectroscopic evidence for weakly-confined two-Majorana spinons in..." which is unclear at this point. (The authors should replace for example 'two-Majorana spinons' simply by 'Majorana fermions from spin fractionization' or similar)

- One page 3 again second paragraph, the authors claim that they observe a "magnetic Raman response not described within a conventional magnon picture" but do not give any reason. The authors should add a few sentences what distinguishes their response from conventional two magnon response.

- On page 7, the authors claim that the magnetic specific heat is related to the Raman susceptibility $C_m \sim X'(w)/w$, which is unclear. E.g. which frequency w is assumed or is it an integral of all frequencies?

This is related to the supplementary material F: (Ref.9 is clearly not the correct one at this point but should be Reiter, PRB 13 1976)

Looking at the papers of Halley 1978 and Reiter 1976 it seems that the specific heat is proportional to the integrated Raman intensity $C_m \sim \text{Int } X'(w)$ and not $C_m \sim X'(w)/w$ This needs to be clarified.

It could explain why the specific heat in Fig.3c does not resemble anything close to the 3D Kitaev spin liquid calculations of the pure Kitaev model Ref.9?

The authors should clarify these important issues!

- Very recently, an article appeared showing that the temperature dependence of the integrated Raman response could provide further evidence for fractionalisation, see arXiv:1602.05277. Did the author look at the T-dependence of the integrated Raman intensity (similar to the RuCl₃ case from Sandiland et al.)?

Reviewer #3 (Remarks to the Author):

I have already reviewed this manuscript twice and witnessed its evolution through the reviewing process. Overall the authors have tried their best to address the comments and criticism of both myself and the first referee.

As I stated previously, the present data could possibly be interpreted as fingerprints of Majorana fermions, but the evidences in terms of selection rules, compositionnal dependence and spectral lineshape, are probably too tenuous to warrant publication in a high profile journal such as Nature Communication. My opinion on this has not changed after reviewing the new version of the ms and the reply to the referees.

It is also visible that the authors did not completely take into account the previous comments and criticisms in their ms. As a results the interpretation of several experimental features is now somewhat unclear.

- For example, as pointed that by the first referee, the low energy continuum is probably dominated by spin waves of the long range magnetic orders, and not primarily from two spinon continuum. The authors seem to acknowledge that in their reply to the referee, but the manuscript does not completely reflects this important distinction. They still state on page 4, that "the magnetic continuum is assigned to two-Majorana spinon excitations".

- Another example is the interpretation of the gapped continuum in gamma compounds. The discussion on page 5 is now completely unclear. Is the gap associated to magnons continuum (as suggested by referee 1) or to anisotropic Kitaev interactions (i.e. fermions continuum) ? This again raises the problem of disentangling the magnons from the two Majorana fermions contributions to the Raman continuum. This question is obviously hard to answer given the lack of structure in the spectral lineshape.

To summarize, the work is of quality and will certainly stimulates further theoretical work on the subject. My feeling is however that it would be better suited to a more specialized journal.

REVIEWERS' COMMENTS:

Reviewer #1 (Remarks to the Author):

The referee now believes the paper is ready for publication with no further comments for the authors

Reviewer #1 (Remarks to the Author):

The authors have addressed most of the points which were raised previously and now carefully state that they 'observe signatures of fractionalization' instead of 'signatures of a quantum spin liquid state'. The manuscript is very timely and contains important new results on both the beta and gamma Li_2IrO_3 which further establish these 3D materials as magnetic materials dominated by the celebrated Kitaev interaction. Their - arguably indirect - evidence for fractionalization will motivate further studies and will receive a lot of attention. The agreement between the clear Raman scattering predictions of Perreault et al. Ref.[8] for the 3D Kitaev spin liquid states is not great but nevertheless several features are surprisingly consistent. In particular the temperature dependence corroborates the authors interpretation in terms of fractionalized particles.

There are additional comments and points which the authors need to clarify, see below. Once these have been addressed, I do recommend publication of this important work in Nature Communication.

Authors) We thank the referee for the positive evaluation of our work as well as for supporting the publication of our manuscript in Nature Communication. In the revised version, we have further improved the presentation style of our manuscript, complying with the referee's comments. Here follows the detailed response.

- On page 3, the authors write "We provide here Raman spectroscopic evidence for weakly-confined two-Majorana spinons in..." which is unclear at this point. (The authors should replace for example 'two-Majorana spinons' simply by 'Majorana fermions from spin fractionization' or similar)

Authors) Strictly speaking, Raman scattering is a second-order process involving a creation or annihilation of a pair of fermions and thus "two-Majorana fermions" are a correct terminology. However, we agree with the referee that "two-Majorana fermions" are a jargon for Raman scattering. For a broad readership, we will replace it by "Majorana fermions" throughout the text, if not unavoidable.

- One page 3 again second paragraph, the authors claim that they observe a "magnetic Raman response not described within a conventional magnon picture" but do not give any reason. The authors should add a few sentences what distinguishes their response from conventional two magnon response.

Authors) Following the referee's suggestion, we state the main findings in a more clear way. In doing that, we stress that the temperature dependence of the integrated Raman intensity obeys Fermi statistics. This clearly distinguishes the fermionic excitation observed in the hyperhoneycomb iridates from the bosonic Raman response expected for conventional magnets.

- On page 7, the authors claim that the magnetic specific heat is related to the Raman susceptibility $C_m \sim \chi(\omega)/\omega$, which is unclear. E.g. which frequency ω is assumed or is it an integral of all frequencies?

This is related to the supplementary material F: (Ref.9 is clearly not the correct one at this point but should be Reiter, PRB 13 1976)

Looking at the papers of Halley 1978 and Reiter 1976 it seems that the specific heat is proportional to

the integrated Raman intensity $C_m \sim \int X'(w)$ and not $C_m \sim X'(w)/w$. This needs to be clarified. It could explain why the specific heat in Fig.3c does not resemble anything close to the 3D Kitaev spin liquid calculations of the pure Kitaev model Ref.9? The authors should clarify these important issues!

Authors) Indeed, the fitting procedure of deducing a magnetic specific heat from the Raman conductivity was vaguely formulated. We clearly state that the magnetic specific heat was evaluated by an integration of $x'(w)/w$ scaled by T . In the previous analysis, the low-frequency contribution was not properly summed and thus the predicted two-peak structure could not be identified. We recalculated the magnetic specific heat by interpolating the Raman conductivity to 0 meV and then integrating $x'(w)/w$. Now the two-peak feature becomes visible at $T_N=0.1$ J and $T^* \sim J$. The higher- T peak is not significantly different from the theoretical value of $T \sim J$ expected for a pure 3D Kitaev system. In addition, the predicted topological transition at $T \sim 0.005$ J is preempted by the long-range order at $T_N=0.1$ J. This is ascribed to the residual interactions. Here the important finding is the robustness of the two-peak structure in the specific heat despite the magnetic order as shown in figure 3d. This corroborates the thermal fractionalization of spins into two types of Majorana fermions in a 3D Kitaev-Heisenberg system. However, we failed to identify a two-step increase of entropy by $(R/2)\ln 2$ (not shown in the manuscript). This may be because the specific heat derived from a Raman response is evaluated on the basis of the hydrodynamic assumption. This kind of arguments was inserted in the new version and the original papers by Halley and Reiter were cited in the main text.

- Very recently, an article appeared showing that the temperature dependence of the integrated Raman response could provide further evidence for fractionalisation, see arXiv:1602.05277. Did the author look at the T -dependence of the integrated Raman intensity (similar to the RuCl_3 case from Sandiland et al.)?

Authors) We thank the referee for spotting this out. We made an in-depth analysis of the temperature dependence of the integrated Raman intensity for the β - and γ - Li_2IrO_3 compounds. The analysis results are now presented in figures 2 c and d. It is evident that the temperature dependence of the Raman spectra is governed by a thermal damping of fractionalized fermionic excitations along with a bosonic background, similar to the case of RuCl_3 . The extracted energy 0.76-0.79 J of the fermions confirms the validity of a fitting procedure. This finite-temperature dynamic behavior lends further support to our original claim that the β - and γ - Li_2IrO compounds harbor the fermionic excitations from spin fractionalization. In the revised version, we added a paragraph to highlight this new finding.

Reviewer #3 (Remarks to the Author):

I have already reviewed this manuscript twice and witnessed its evolution through the reviewing process. Overall the authors have tried their best to address the comments and criticism of both myself and the first referee.

As I stated previously, the present data could possibly be interpreted as fingerprints of Majorana fermions, but the evidences in terms of selection rules, compositionnal dependence and spectral lineshape, are probably too tenuous to warrant publication in a high profile journal such as Nature Communication. My opinion on this has not changed after reviewing the new version of the ms and the reply to the referees.

It is also visible that the authors did not completely take into account the previous comments and criticisms in their ms. As a results the interpretation of several experimental features is now somewhat unclear.

Authors) We appreciate a series of valuable comments by the referee which led to a more complete formulation of our findings. In course of revising the manuscript, a number of theoretical works have been published on a finite-temperature signature of fractionalized fermionic excitations in a Kitaev spin system [J. Nasu et al., arXiv:1602.05299 (2016)]. In light of this theoretical framework, we provide more compelling evidence for our original assertion.

We agree with the referee that the strictest proof of Majorana fermions is given in terms of the spectral form and selection rule of the spinon continuum. However, we would like to recall that a finite-temperature dynamical Raman response allows achieving this goal in a Kitaev magnet within a well-established theory. Indeed, we find that the temperature dependence of an integrated Raman intensity follows the fermionic form consistent with the recent theory. Now we believe that our findings are scientifically based on a solid ground and are a significant achievement in this rapidly advancing area. As the two referees recognized, the observation of fermionic excitations in 3D Kitaev-Heisenberg compounds exerts a strong impact on the broad community, deserving publications in a high profile journal rather than in a specialized journal.

- For example, as pointed that by the first referee, the low energy continuum is probably dominated by spin waves of the long range magnetic orders, and not primarily from two spinon continuum. The authors seem to acknowledge that in their reply to the referee, but the manuscript does not completely reflects this important distinction. They still state on page 4, that "the magnetic continuum is assigned to two-Majorana spinon excitations".

Authors) As the referee rightly pointed out, the low-energy magnetic continuum is given by correlated magnons while the high-energy part by spinon continuum. Thus, the sentence "the magnetic continuum is assigned to two-Majorana spinon excitations" misleads the reader. We rephrase the above sentence with "The magnetic continuum arises mainly from two-Majorana spinon excitations".

- Another example is the interpretation of the gapped continuum in gamma compounds. The discussion on page 5 is now completely unclear. Is the gap associated to magnons continuum (as suggested by referee 1) or to anisotropic Kitaev interactions (i.e. fermions continuum) ? This again raises the problem of disentangling the magnons from the two Majorana fermions contributions to the Raman continuum. This question is obviously hard to answer given the lack of structure in the spectral lineshape.

Authors) As discussed above, the low-energy magnetic excitations are correlated magnons. In the low-energy regime, thus disentangling the magnons from the spinons is not a well-defined question. The only thing we can say is that the unusually huge gap of spin waves in gamma-Li₂IrO₃ is due to anisotropic Kitaev interactions. The existence of a sizable energy gap is not observed in beta-Li₂IrO₃ with nearly isotropic Kitaev interactions.

To summarize, the work is of quality and will certainly stimulates further theoretical work on the subject. My feeling is however that it would be better suited to a more specialized journal.

Authors) As all referees admitted, the physical identification of Majorana fermions is one of the most intensely pursued goals in current experimental condensed matter physics. Our observation of fermionic excitations in beta- and gamma-Li₂IrO₃ represents a scientific breakthrough on a scale that warrants publication in Nature Communications. It surely will be of interest to a wide audience in the condensed matter physics community.